# Prevalence of Avascular Necrosis Following Surgical Treatments in Unstable Slipped Capital Femoral Epiphysis (SCFE): A Systematic Review and Meta-Analysis

**DOI:** 10.3390/children9091374

**Published:** 2022-09-11

**Authors:** Vijayanagan Veramuthu, Ismail Munajat, Md Asiful Islam, Emil Fazliq Mohd, Abdul Razak Sulaiman

**Affiliations:** 1Department of Orthopaedics, School of Medical Sciences, Universiti Sains Malaysia, Kubang Kerian 16150, Kelantan, Malaysia; 2Department of Haematology, School of Medical Sciences, Universiti Sains Malaysia, Kubang Kerian 16150, Kelantan, Malaysia; 3Institute of Metabolism and Systems Research, University of Birmingham, Birmingham B15 2TT, UK

**Keywords:** unstable slipped capital femoral epiphysis, unstable slip, avascular necrosis, osteonecrosis

## Abstract

The choice of treatment for unstable and severely displaced slipped capital femoral epiphysis (SCFE) is controversial. This meta-analysis was conducted to determine the prevalence of femoral head avascular necrosis (AVN) following various treatments for unstable SCFE. Various databases were searched to identify articles published until 4 February 2022. A random-effects model was used to examine prevalence as well as risk ratios with confidence intervals (CIs) of 95%. Thirty-three articles were analyzed in this study. The pooled prevalences of AVN in pinning in situ, pinning following intentional closed reduction, pinning following unintentional closed reduction, and open reduction via the Parsch method, subcapital osteotomy and the modified Dunn procedure were 18.5%, 23.0%, 27.6%, 9.9%, 18.6% and 19.9%, respectively. The risk of developing AVN in pinning following intentional closed reduction was found to be 1.62 times higher than pinning in situ; however, this result was not significant. The prevalence of AVN in open reduction was lowest when performed via the Parsch method; however, this finding should be interpreted with caution, since the majority of slips so-treated are of mild and moderate types as compared with the subcapital osteotomy and modified Dunn procedures, which are predominantly used to treat severely displaced slips. As the risk ratio between intentional closed reduction and the modified Dunn method showed no significant difference, we believe that the modified Dunn method has the advantage of meticulously preserving periosteal blood flow to the epiphysis, thus minimizing AVN risk. In comparison with intentional closed reduction, the modified Dunn method is used predominantly in cases of severe slips.

## 1. Introduction

SCFE is a frequent hip issue that typically develops during early teenage years [1]. SCFE causes the femoral neck to be in varus and externally rotated because the femoral head remains in the acetabulum as the neck moves to the front and rotates externally [2]. SCFE is divided into two categories according to Loder’s classification: (i) stable and (ii) unstable [3]. A child who can walk with or without crutches is deemed stable, while one who cannot bear their weight even with crutches is termed unstable, regardless of the length of the symptom [4]. Overweight, retroverted femoral head, growth spurts and excessive physeal obliquity are some of the factors that contribute to the development of slip [5,6].

The aetiology of avascular necrosis (AVN) following SCFE has a complex pathogenesis [7]. Factors leading to the development of AVN include (i) the stability and severity of the slip, (ii) the method of reduction, (iii) the type of surgical procedure, (iv) the timing of surgery and (v) the timing of presentation [8]. The primary cause of osteonecrosis is attributed to the femoral neck being acutely displaced from the epiphysis, which reduces perfusion of the posterior retinacula vessel to the epiphysis [9].

Loder et al. [4] first proposed SCFE instability in 1993 after they found a substantial link between poor outcome and instability. This study observed that teenagers with unstable slip were at a higher risk of femoral head AVN than those with stable slip. Other investigators replicated similar findings, demonstrating that unstable slips were associated with a greater incidence of AVN, ranging between 10 and 60% [5,10,11,12,13,14].

According to the majority of clinicians, surgical treatment is recommended when an unstable slip is diagnosed. The treatment goals of unstable SCFE include prevention of further slippage, avoidance of osteonecrosis and prevention of future impingement. Various methods have been reported, including (i) pinning in situ, (ii) pinning following unintentional closed reduction, (iii) pinning following intentional closed reduction and (iv) open reduction via subcapital correction osteotomy, the modified Dunn method and the open Parsch method [15,16,17,18,19].

Pinning in situ percutaneously with one screw is proven to have the lowest osteonecrosis risk in stable slip [20,21]. However, in unstable slip, the best treatment for achieving the lowest AVN risk is still undetermined [22,23]. Furthermore, pinning with a percutaneous screw following closed reduction has been previously discouraged, as it is claimed that the pre-reduction maneuver increases the risk of femoral head osteonecrosis [14,24]. However, the open subcapital realignment surgery performed by surgically dislocating the hip, as in the modified Dunn method, has rapidly gained in popularity in treating severely unstable slips. It is claimed that the modified Dunn method can be used to realign severe slips without increasing the rate of osteonecrosis [18,25].

Up to now, there has been no meta-analysis that has compared the outcomes of various surgical procedures, especially AVN risk. Therefore, this study aimed to determine the prevalence of femoral head AVN following various surgical treatments in unstable SCFE.

## 2. Materials and Methods

### 2.1. Guidelines and Protocol

To investigate the prevalence of femoral head AVN following management of unstable SCFE, a systematic review and meta-analysis was undertaken according to the PRISMA guidelines. This research was registered with PROSPERO under the number CRD42020212718.

### 2.2. Data Sources and Searches

Studies published before 4 February 2022 were identified in the PubMed, Google Scholar, Scopus and Cochrane Library databases, with no language limitations. Appendix A includes all the details of the search strategy. Case reports, editorials, comments and review articles were excluded. In order to establish a thorough search method, the references of the included papers were examined. To eliminate similar studies, EndNote X8 software was employed.

### 2.3. Study Selection

To find studies that were eligible for inclusion in the analysis, the authors (VV and IM) independently reviewed the titles and abstracts of the records they found, followed by a full-text review. With the help of a third author (MAI), disagreements concerning inclusion were explored and resolved by consensus. If a study was cross-sectional or cohort in nature, it was considered eligible for inclusion. Adolescents of both sexes fulfilling the criteria of Loder et al. [4] were considered eligible participants. In this study, only individuals who were classified as unstable SCFE patients were included.

### 2.4. Data Extraction and Quality Assessment

From the finally selected articles, three authors (VV, IM and MAI) independently extracted data of interest in Excel sheets. The extracted data included: (i) last name of the first author, (ii) the year the article was published, (iii) study design, (iv) data collection period, (v) number of participants, (vi) total number of hips, (vii) number of AVN cases, (viii) mean age of the participants, (ix) type of intervention, (x) country of origin and (xi) mean follow-up duration. Data on the severity of slips were also captured. Utilizing the Joanna Briggs Institute (JBI) critical appraisal tools, three authors (VV, IM and MAI) made an independent assessment of the quality of the included studies [26]. Appendix A revealed the quality assessment of the included studies. The articles were regarded as being of low quality (high risk of bias), moderate quality (moderate risk of bias), or high quality (low-risk of bias) when overall scores were ≤49, 50–69 or ≥70%, respectively [27].

### 2.5. Data Synthesis and Analysis

The prevalence and 95% CIs of femoral head AVN among individuals undergoing therapy for unstable SCFE were calculated. The means and standard deviations for continuous data were calculated, and the risk of developing AVN following any treatment was revealed as a risk ratio (RR) with a 95% CI for dichotomous data. A random-effects model was used for the calculations in all analyses. The *I*^2^ statistic was used to determine study heterogeneity (*I*^2^ > 75% showed considerable heterogeneity) and the significance level was presented according to Cochran’s Q test. As subgroups, the prevalences of femoral head AVN among different types of surgical treatment were analyzed. In order to test the results’ robustness, sensitivity analyses were carried out using methods such as (i) excluding low- or moderate-quality studies, (ii) excluding outlier studies and (iii) considering only cross-sectional studies. Prevalence estimates were plotted against standard errors in a funnel plot to measure publication bias, and Egger’s test was used to validate funnel plot asymmetry. Outlier studies and potential sources of heterogeneity were detected by constructing a Galbraith plot. Metaprop codes in meta (version 4.15-1) were used to generate the analyses and plots. The Metafor (version 2.4-0) packages of R (version 3.6.3) in RStudio (version 1.3.1093) were utilized. Review Manager 5.4 was also used to calculate the risk ratio (RR) of AVN when comparing the data between two different surgical interventions.

## 3. Results

### 3.1. Study Selection

Our preliminary search turned up 737 articles, from which 158 studies were excluded because they consisted of review articles, case reports, editorials, comments and duplicate studies. For eligibility, 579 papers were reviewed based on titles and abstracts, with 546 being removed due to failure to fulfil the inclusion criteria. In the end, the systematic review and meta-analysis included 33 studies (Figure 1).

### 3.2. Study Characteristics

In Table 1, the included studies’ detailed features and references are listed. This meta-analysis summarizes findings from 33 studies, including 858 hips undergoing different surgical procedures for unstable slips. The included studies were published from 2001 to 4 February 2022 and comprise 26 cross-sectional and 7 cohort studies. The surgical procedures included (i) pinning in situ (PIS), (ii) pinning following intentional closed reduction, (iii) pinning following unintentional closed reduction, (iv) open reduction (OR) via the Parsch method, (v) OR via subcapital osteotomy (SCO) using the anterior/anterolateral approach, including Fish, original Dunn and cuneiform osteotomy methods, and (vi) OR through surgical hip dislocation (modified Dunn). The mean age of surgical intervention was 11.6–14.0 years old. The time taken from initial admission to surgical theatre entry ranged from 3 h to 30 days. The follow-up duration included in this meta-analysis ranged from 9 months to 17 years. Most fixation procedures use cannulated screws rather than wires. Subcapital osteotomy and modified Dunn involve shortening of the neck, which is not the case in open reduction via the Parsch method.

**Table 1 children-09-01374-t001:** Major characteristics of the included studies.

Study	Type of Study Conducted, Country	Time Period of Intervention ± SD	Total Participants (Female)	Mean Age (Years) ± SD	Type of Intervention	Reduction	Mean Duration Follow-Up
Alshryda 2014 [11]	Cross-sectional study, UK	Immediate	22 (10)	13.4	Pinning in situ; Subcapital osteotomy (Fish)	No, in in situ; Yes, in open reduction; No severity assessed	10 years
Alves 2012 [13]	Cross-sectional study, Canada	OS: 22.16 h, ± 7.86 CM: 24.25 h, ± 7.86	6 (3) 6 (3)	12.5 ± 1.4 11.8 ± 1.9	Pinning following closed reduction; Modified Dunn	Yes, in closed reduction and in modified Dunn	3–4 years
Bali 2015 [28]	Cross-sectional study, UK	9.4 days (2–42 days)	34 (14)	13.1 (11–16)	Modified Dunn	Yes	54 months (15–102 months)
Chen 2009 [17]	Cross-sectional study, USA	28.4 h ± 26	29 (10)	11.6 ± 2	Pinning following unintentional closed reduction; Open reduction without neck shortening in 5 cases (Parsch method)	Yes, in unintentional closed and open reduction	5.5 years (2–11.2 years)
Cosma 2016 [29]	Cross-sectional study, Romania	Not stated	10 (7)	12.7	Pinning in situ; Modified Dunn	No, in in situ; Yes, in Dunn	18 months
Davis 2017 [30]	Cross-sectional study, USA	13.9 h (2.2–23.4 h)	Not specified	12.5	Modified Dunn	Yes	27.9 months
Herngren 2018 [31]	Cohort study, Sweden	Not stated	61 patients	11.1–14.9	Pinning in situ; Pinning following closed reduction; Subcapital osteotomy; Open reduction (Parsch method)	No, in in situ; Yes, in closed and open reduction; Severity assessed but proportion not calculated; 32 cases had intentional CR 5 cases had unintentional CR	36 months
Ilharreborde 2016 [32]	Cross-sectional study, France	1–30 days	82 (35)	13 ± 2	Subcapital shortening osteotomy (cuneiform)	Yes	28 months ± 8
Jackson 2016 [33]	Cohort study, USA	11.7 h (3–22 h)	9 (4)	14 (9–15)	Modified Dunn	Yes	9–29 months
Masquijo 2017 [34]	Retrospective cohort, Argentina	Not stated	20 (10)	12 (10–16)	Modified Dunn	Yes	40.4 months (12–84 months)
Kitano 2015 [35]	Cross-sectional study, Japan	<24 h–7 days	21 (7)	12.1 (10.7–14.5)	Pinning following closed reduction; Pinning in situ	7 preoperative tractions in closed reduction group; No severity assessed; 14 cases had intentional CR	Not stated
Kohno 2016 [36]	Cross-sectional study, Japan	<24 h = 9 24 h–7 days = 3 >7 days = 12	60 (21)	11.8 ± 1.8 11.8 ± 1.7	Pinning in situ; Pinning following closed reduction	No, in in situ; Yes, in closed reduction; Severity assessed but using posterior tilting angle (PTA); Mean PTA 60 in closed reduction; Mean PTA 47 in PIS; 43 cases had intentional closed reduction; 17 cases had PIS	4.7 years (1.0–14.5 years)
Lang 2019 [37]	Retrospective cohort, USA	Immediate	18 (5)	11.7	Pinning in situ	No	31.2 months (12–62 months)
Lerch 2019 [38]	Retrospective case series analysis, Switzerland	Not stated	14 hips	13 ± 2, (9–19)	Modified Dunn	Yes	9 years ± 4 (2–17 years)
Madan 2013 [39]	Prospective study, UK	Not stated	17 hips	12.9 (10–20)	Modified Dunn	Yes	38.6 months (24–84 months)
Masse 2012 [40]	Cross-sectional study, Italy	Not stated	2 hips	13.5 in boys 12 in girls	Modified Dunn procedure with extended retinacula flap	Yes	24 months
Mulgrew 2011 [41]	Cross-sectional study, UK	Not stated	10 (6)	12.6	Pinning in situ	No	17.8 months
Ng 2019 [42]	Cross-sectional study, Singapore	57.7 h	23 (6)	11.9	Pinning in situ; Manipulation, reduction and screw fixation (*n* = 5)	No, in in situ; Yes, in manipulative group; No severity assessed; 5 cases had intentional closed reduction; 18 cases had PIS	23 months
Nortje 2009 [43]	Cross-sectional study, South Africa	Not stated	Group B 20 unstable hips	Group B 13.6 (9–16)	Group B Single screw fixation in situ	No	2 years
Palocaren 2010 [44]	Cohort study, USA	61 h ± 70.2	27 patients	12.2 ± 1.58	Pinning in situ	No	3.1 ± 1.9 years
Parsch 2009 [45]	Cohort study, Germany	<24 h	64 (27)	8–16	Open reduction; No shortening; Smooth K wire fixation (Parsch method)	Yes	4.9 years (18–104 months)
Persinger 2016 [46]	Cross-sectional study, USA	13.9 h (2.17–23.4 h)	30 (15)	12.37 (8.75–14.8)	Modified Dunn	Yes	29.3 months (12–82 months)
Phillips 2001 [47]	Cross-sectional study, UK	<24 h	14 (5)	13	Crawford–Adams pin, 10; Cannulated screw, 1; Smith Peterson nail, 1; Dunn osteotomy, 2	Gentle manipulative closed reduction; Open reduction	2 years
Rached 2012 [48]	Cross-sectional study, Brazil	Not stated	26 (10)	13 (8.2–17.2)	Steinmann pin/single cannulated screw; Multiple pin	Closed reduction and fixation	2 years
Sankar 2010 [49]	Cross-sectional study, USA	Not stated	14 patients	12.6 (6.5–17.8)	Pinning in situ; Closed reduction; Open reduction (8 modified Dunn, 8 open reduction Parsch method)	No, in in situ; Yes, in closed and open reduction; No severity assessed	3.2 years (1–10 years)
Seller 2006 [50]	Cross-sectional study, Germany	Not stated	29 patients	11–16	Closed reduction and k-wire fixation	Yes	3.5 years
Slongo 2010 [51]	Cross-sectional study, Switzerland	Not stated	3 hips	11.9 ± 2.02 Boys 12.5 Girls 10.8	Modified Dunn	Yes	24 months (23–62 months)
Souder 2014 [52]	Cohort study, USA	Not stated	14 hips	12.2 ± 1.6 (9.3–16.7)	Pinning in situ; Modified Dunn	No, in in situ; Yes, in modified Dunn	Dunn procedure 15.6 ± 7 months In situ pinning 31.4 ± 22.2 months
Ulici 2017 [53]	Cross-sectional study, Romania	Not stated	15 patients	12.5 (9–16)	Pinning	No, in in situ	30 months
Upsani 2014 [54]	Cross-sectional study, USA	Not stated	26 patients	12.6 boys (11–16) 11.4 girls (9–17)	Modified Dunn	Yes	2.6 years (1–8 years)
Vanhegan 2015 [55]	Cross-sectional study, UK	Delayed	57 (22)	13.1 (9.6–20.3)	Subcapital osteotomy (cuneiform)	Yes	7 years (2.8–13.9 years)
Walton 2015 [19]	Cross-sectional study, UK	Not stated	45 patients	12.6 (10–14) 12.6 (9–14)	Closed unintentional reduction; Subcapital osteotomy (cuneiform)	Yes, in closed unintentional reduction and in subcapital osteotomy	28 months (11–48 months) 30 months (10–50 months)
Zang 2018 [56]	Cross-sectional study, Japan	Not stated	3 unstable hips	11.8 (8–14)	Subcapital osteotomy (cuneiform)	Yes	4.5 years (1.5–9.9 years)

### 3.3. Outcomes

Overall, the pooled prevalences of osteonecrosis in unstable SCFE patients following closed pinning and open reduction were 21.9% (95% CI: 17.5–27.2%) and 18.5% (95% CI: 12.6–26.3%), respectively (Figure 2). The risk of developing AVN following closed pinning compared to open reduction in patients with unstable SCFE was not significant (RR: 1.06; 95% CI: 0.50–2.28) (Figure 2).

Closed pinning: pinning in situ + pinning following closed reduction. Open reduction: open reduction via the Parsch method + subcapital osteotomy via the anterior or anterolateral approach + the modified Dunn procedure using surgical hip dislocation.

In subgroup analyses, the prevalences of AVN in pinning in situ, pinning following intentional closed reduction, pinning following unintentional closed reduction, pinning following intentional and unintentional closed reduction, and open reduction via the Parsch method, subcapital osteotomy and modified Dunn procedures were 18.5% (95% CI: 13.3–25.1%), 23.0% (95% CI: 15.3–33.0%), 27.6% (CI: 12.2–51.0%), 24.2% (CI: 17.2–33.0%), 9.9% (95% CI: 3.2–27.0%), 18.9% (95% CI: 11.7–29.1%) and 19.9% (95% CI: 11.7–31.8%), respectively (Table 2 and Appendix A).

Severity was assessed in 20 out of 33 studies using the Southwick angle in unstable SCFE, and one study used posterior tilting angle for assessment. Since open reduction via the Parsch method showed the lowest prevalence of AVN, slip severity was studied and it was found that the percentages of mild, moderate and severe slips treated via the Parsch method were 31.2%, 37.5% and 31.2%, respectively (Table 3). However, these percentages were mainly derived from a single study by Parsch 2009 [45].

The rate of AVN in pinning in situ was 19.2% in mild cases, 19.0% in moderate cases and 19.2% in severe cases (Table 4). Meanwhile, the rate of AVN in pinning following intentional closed reduction was 15.7% in mild cases, 16.3% in moderate cases and 25.0% in severe cases (Table 4). The slips treated with intentional closed reduction were mainly moderate (53%) and severe (37%) slips. In contrast, for pinning in situ, the proportions were fairly even across all severity grades. The rate of AVN according to severity in pinning following an unintentional reduction was unable to be analyzed in view of no severity information being documented in two publications by Souder et al. 2014 [52] and Palocaren et al. 2010 [44] (Table 4).

The risk of developing AVN in pinning following intentional closed reduction was 1.62 times higher than in pinning in situ in patients with unstable slip; however, this result was not significant (Figure 3A). The risk ratios for developing avascular necrosis following pinning in situ versus the Parsch method of open reduction, subcapital osteotomy and modified Dunn were 0.67, 0.70 and 1.81, respectively, otherwise the results were not significant (Figure 3B–D).

The risk ratios for developing avascular necrosis following intentional closed reduction compared to the Parsch method of open reduction and modified Dunn were 0.83 and 1.21, respectively (Figure 3E,G). However, these results were not significant. Meanwhile, the risk of osteonecrosis in intentional closed reduction compared to open reduction via subcapital osteotomy was notably lower. The risk ratio was 0.28, which was statistically significant (Figure 3F); however, only one study was documented in that analysis. The cases in which modified Dunn and subcapital osteotomy were performed were predominately severe slips, the percentages for which were 88.9% and 99.1%, respectively (Table 5). In contrast, for the Parsch method, the percentage of severe slips treated was only 31.0% (Table 5).

### 3.4. Quality Assessment and Publication Bias

The quality assessment in this meta-analysis showed that 36.4%, 60.6% and 3.0% of studies were of high, moderate and low quality, respectively (Appendix A). The findings obtained with funnel plots as well as Egger’s test revealed no publication bias (*p*-value > 0.05) in estimating the prevalence of AVN in unstable SCFE following closed pinning and open reduction (Figure 4).

### 3.5. Sensitivity Analyses

Sensitivity analyses assessing the prevalence of AVN following closed pinning and open reduction in SCFE patients showed very marginal differences (5.9% lower to 2.6% higher) in overall pooled prevalence compared with the main findings (Table 6 and Appendix A). Therefore, according to the above analyses, the findings of the current study are strong and trustworthy. From the Galbraith plots, no study was identified as a potential source of heterogeneity in the closed pinning group, while only one study was identified as such in the open reduction group (Figure 5).

## 4. Discussion

The stability of a slip, as documented previously by Loder, is the most useful piece of information as it can be used to prognosticate AVN risk for the femoral head, which is a significant complication leading to poor outcomes. Out of 55 patients who had unstable SCFE, Loder demonstrated in his series that 47% went on to develop AVN while none of the patients with stable hips did [4]. Therefore, once SCFE is diagnosed, surgery is needed to stabilize the epiphysis, prevent further slipping and minimize AVN risk. In addition, an acutely slipped epiphysis from the neck may kink or tear the retinacular artery, the main artery perfusing the epiphysis running at the femoral neck posterosuperiorly.

Even though earlier meta-analyses [57,58,59,60] acknowledged that AVN was a common complication among patients with unstable SCFE, subgroup analyses were not thoroughly conducted. Therefore, in our meta-analysis, we performed a subgroup analysis of AVN risk in PIS versus CR, PIS versus different types of open reduction (Parsch method, SCO and modified Dunn) and CR versus different types of open reduction (Parsch method, SCO and modified Dunn). In our analyses, PIS and CR were considered closed pinning methods since the slips are not surgically opened during the intervention, whereas the Parsch method, SCO and modified Dunn were regarded as open reduction methods since the femoral head is, indeed, opened during slip reduction.

There is no solid consensus on the surgical treatment for unstable slips with regard to the lowest risk of AVN [23]. Our systematic review and meta-analysis (SRMA) revealed that, following closed pinning and open reduction, the pooled prevalences of AVN were 21.9% and 18.5%, respectively. In our comparative analysis, the risk of AVN between closed pinning and open reduction was not statistically significant. This comparison gives a rough idea of the closed and open methods of unstable slip fixation.

Robust evidence of outcomes following various surgical options is essential for identifying best treatments. The moderate and severe forms of unstable slips present challenges regarding whether we should pin in situ or perform closed intentional reduction prior to pinning. For closed pinning, we performed a subgroup analysis of pinning in situ versus pinning following intentional closed reduction. The pooled prevalence of osteonecrosis was lower (18.5%) in pinning in situ than pinning following intentional closed reduction (23.0%). Even though the risk ratio analysis between intentional closed reduction and pinning in situ was statistically not significant, the risk of AVN was 1.62 times higher in closed intentional reduction. There has been no study comparing pinning in situ versus unintentional closed reduction for inclusion in this analysis. There has also been no study comparing intentional versus unintentional closed reduction. Our finding was consistent with the earlier meta-analysis by Lowndes et al. 2009 [57], in which they were unable to ultimately decide whether there was a significant difference in AVN risk between reduction and non-reduction groups. The temptation to realign the slipped epiphysis to its original position is always offset by the concern about osteonecrosis, which is devastating in the long term [16,61,62]. Comparing these two methods and in view of the higher risk ratio for AVN in intentional closed reduction, we believe that pinning in situ is a safer option in unstable slips. In our analysis of intentional closed reduction, there is an increasing trend of AVN as severity increases, since more moderate and severe slips are found to undergo intentional closed reduction, as shown in Table 4. However, in PIS, the number of cases is almost equally distributed among severity grades (Table 4).

From this meta-analysis, as shown in Table 2, the prevalence of AVN in cases of treatment via the Parsch method was lowest compared to subcapital osteotomy and modified Dunn methods. However, since the majority of slips treated with the Parsch method were mild and moderate, while subcapital osteotomy and modified Dunn procedures were predominantly used to treat severely displaced slips, as shown in Table 5, this result should be interpreted with caution. More complex surgical procedures are needed in more severe SCFE cases, and therefore AVN cannot be correlated with the procedure itself; the stage of presentation of the pathology has to be considered as the risk factor.

Managing a high-grade unstable slip is rather challenging, as pinning in situ right at the center of the epiphysis may be technically impossible [23]. This situation obtains in cases of severe slip when the epiphysis is completely displaced to the posterior part of the neck. In this instance, pinning in situ of the epiphysis is no longer feasible without penetrating the posterior part of the neck, putting already compromised retinacular vessels at greater risk. Regarding severely displaced slips, the dispute concerning which type of reduction to perform (closed or open) is never-ending. The AVN risk between closed reduction and SCO was shown to be significant; however, only a study by Herngren et al. [31] provided data for this aspect of the analysis and we need to be careful in interpreting the result with such a very small number of studies. Thus, further studies will be required to verify the finding.

The risk ratio for AVN with intentional closed reduction was 1.21 times higher than with the modified Dunn method, even though no significant difference was noted. In severely displaced slips, we believe the modified Dunn method to be a wiser choice than intentional closed reduction based on this risk ratio. The modified Dunn method has the benefit of meticulously preserving the periosteal blood flow to the epiphysis during surgical reduction. The possibly kinked or severed periosteal blood supply is not addressed and managed wisely with intentional closed reduction, as it is in the Dunn method, before fixation. Furthermore, one of the most important findings of our analysis is that modified Dunn was predominantly chosen to deal with severe slips, the rate being as high as 88.9%, compared to intentional closed reduction, the rate of which was only 37%. Since the modified Dunn procedure is able to reduce a slip completely, the future risk of femoroacetabular impingement is no longer an issue, provided that the head is free from osteonecrosis.

Dunn [63] subcapital corrective osteotomy combined with the open surgical hip dislocation popularized by Ganz [64] is known as the modified Dunn method [25], which is, however, more invasive than pinning in situ and reserved for patients with severe slips. This technique has been a subject of interest, since it has been thought to minimize AVN risk by directly preserving the retinacular vessels perfusing the femoral epiphysis [15,25,65,66]. The periosteal sleeve of the neck has to be dissected to create a flap [18]; thus, it is still doubtful whether this surgery would also increase AVN risk. Periosteal dissection and neck osteotomy might endanger the susceptible retinacular arteries nearby. The surgeon’s skill may be a factor contributing to the osteonecrosis rate associated with the modified Dunn technique [22]. Loder and Dietz were unable to ascertain whether the modified Dunn procedure was superior to other interventions [23]. The procedure itself is challenging, technically demanding and prone to complications, even when performed by an experienced surgeon.

The vessels supplying the epiphysis might be disturbed during an initial slip if no reduction is performed. This theory was supported by Alves et al. [13], who discovered no connecting periosteum from the neck to the slipped epiphysis in one of his cases subjected to surgical hip dislocation, making the risk of AVN almost 100%.

In terms of the strength of this meta-analysis, it is the first study to compare the risk of femoral head AVN for various surgical interventions of unstable SCFE. The risk of femoral head AVN for pinning in situ and for pinning following closed reduction, as revealed in our meta-analysis, has not been assessed elsewhere. In our meta-analysis, the sensitivity analysis results and the principal results were identical, suggesting that the meta-analysis results were trustworthy. Furthermore, the quality assessments showed that high methodological quality (low risk of bias, with a score of more than 50%) was documented in 81.8% (27 out of 33) of the studies included in this meta-analysis, reflecting the reliability of the results. Neither the funnel plots nor the Egger’s tests for closed pinning and open reduction showed publication bias, again supporting the reliability of the results.

The current meta-analysis has a few limitations. The comparative meta-analysis within the subgroups was conducted with a small number of studies; thus, the results might not be fully representative and should be interpreted with caution. A considerable amount of heterogeneity, as evidenced by *I*² statistics, was present in this meta-analysis. Despite utilizing Galbraith plots to detect the causes of heterogeneity, the real causes of heterogeneity might not lie within the analyses. Finally, most of the comparative analyses of surgical interventions in this study showed no significant differences, so future analyses including more studies are recommended.

## 5. Conclusions

Since the overall prevalence of AVN is lower in pinning in situ than in closed reduction, we believe pinning in situ to be the preferable surgical treatment for slips that are mildly and moderately displaced. Intentional reduction by the closed method and severity of slip are risk factors for AVN. Based on this meta-analysis, any intention to perform a closed reduction for an unstable slip should be weighed and considered carefully, since the risk for osteonecrosis is higher. Regarding open reduction, the prevalence of AVN for the Parsch method was found to be the lowest compared to subcapital osteotomy and the modified Dunn method. However, since most of the slips treated with the Parsch method were mild and moderate, while those treated with subcapital osteotomy and modified Dunn procedures were predominantly severely displaced slips, this finding should be interpreted with caution. Since the risk ratio for intentional closed reduction and the modified Dunn method showed no significant difference, we believe that modified Dunn has the advantage of meticulously preserving periosteal blood flow to the epiphysis, thus minimizing AVN risk. In comparison with intentional closed reduction, modified Dunn is predominantly used to treat severe slips.

## Figures and Tables

**Figure 1 children-09-01374-f001:**
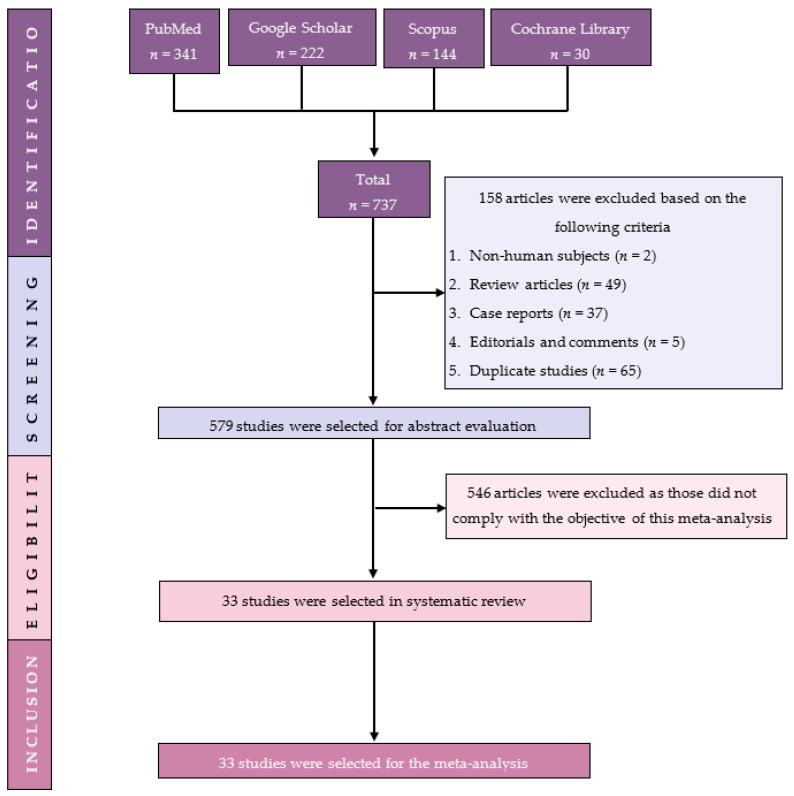
PRISMA flow diagram of study selection.

**Figure 2 children-09-01374-f002:**
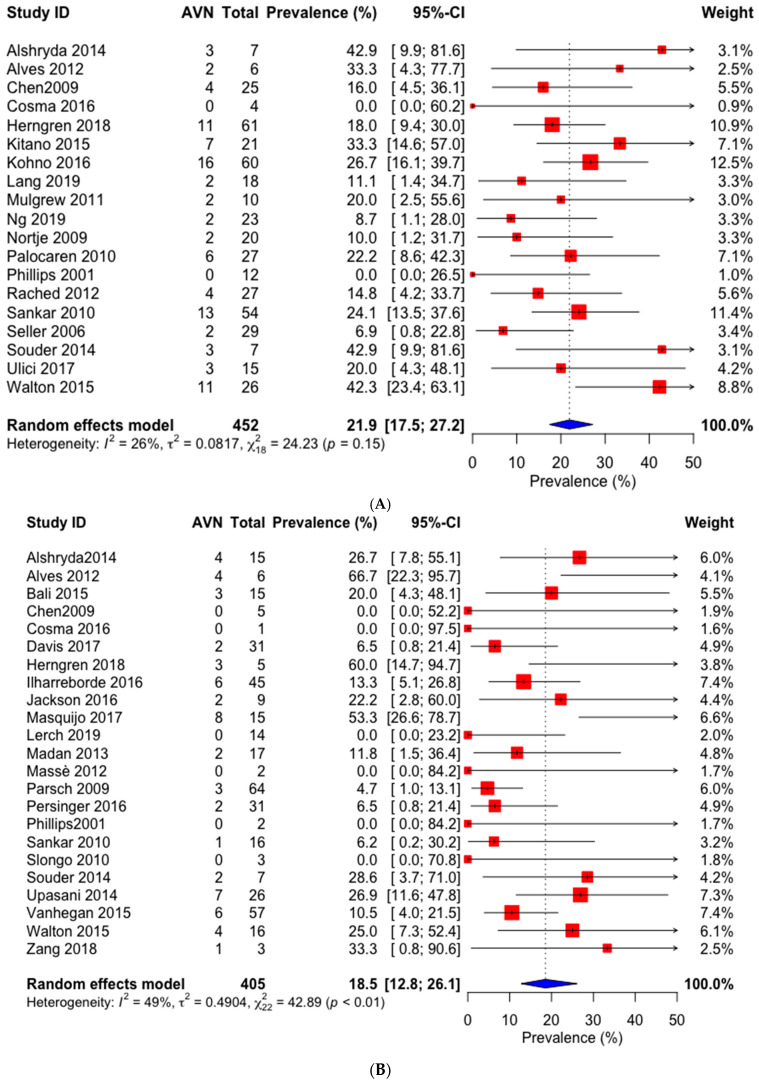
Prevalence (**A**,**B**) and risk (**C**) of developing avascular necrosis following closed pinning and open reduction in patients with unstable slipped capital femoral epiphysis. (**A**) Closed pinning. (**B**) Open reduction. (**C**) Estimated risk ratio in closed pinning versus open reduction.

**Figure 3 children-09-01374-f003:**
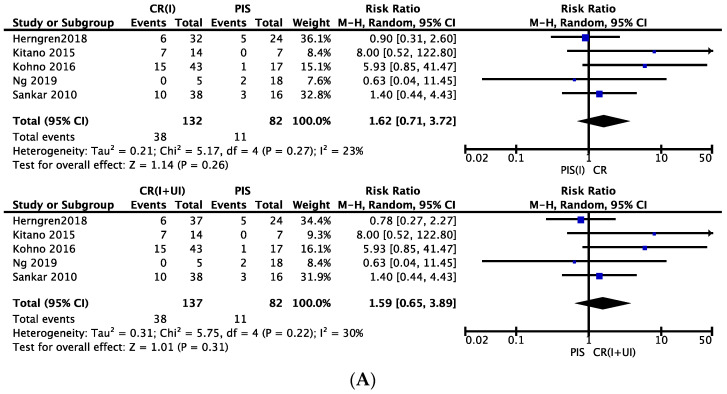
Subgroup analyses of the estimated risk ratios. PIS = pinning in situ, CR(I) = pinning following intentional closed reduction, CR(I+UI) = pinning following intentional + unintentional closed reduction, OR = open reduction, SCO = subcapital osteotomy via anterior/anterolateral hip approach. (**A**) Estimated risk ratio in CR(I) and CR(I+UI) versus PIS. (**B**) Estimated risk ratio in PIS versus OR (Parsch method). (**C**) Estimated risk ratio in PIS versus OR (subcapital osteotomy (SCO)). (**D**) Estimated risk ratio in PIS versus OR (modified Dunn). (**E**) Estimated risk ratio in CR(I) and CR(I+UI) versus OR (Parsch). (**F**) Estimated risk ratio in CR(I) and CR(I+UI) versus OR (SCO). (**G**) Estimated risk ratio in CR(I) versus OR (modified Dunn).

**Figure 4 children-09-01374-f004:**
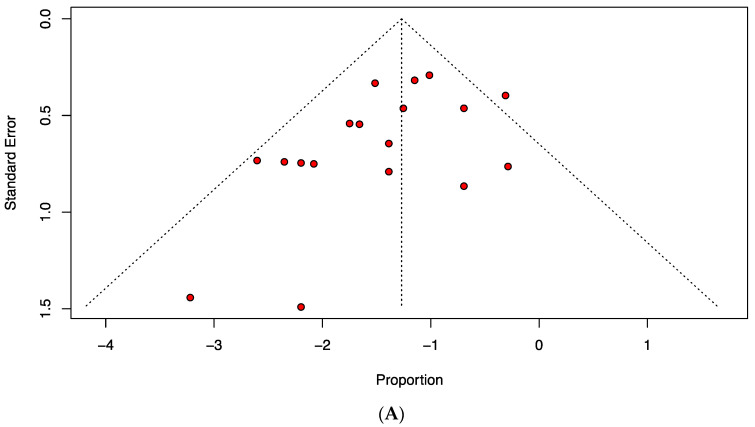
Funnel plots representing publication bias in relation to the prevalence of avascular necrosis following (**A**) closed pinning and (**B**) open reduction in patients with unstable slipped capital femoral epiphysis. (**A**) Egger’s test *p* = 0.102. (**B**) Egger’s test *p* = 0.882.

**Figure 5 children-09-01374-f005:**
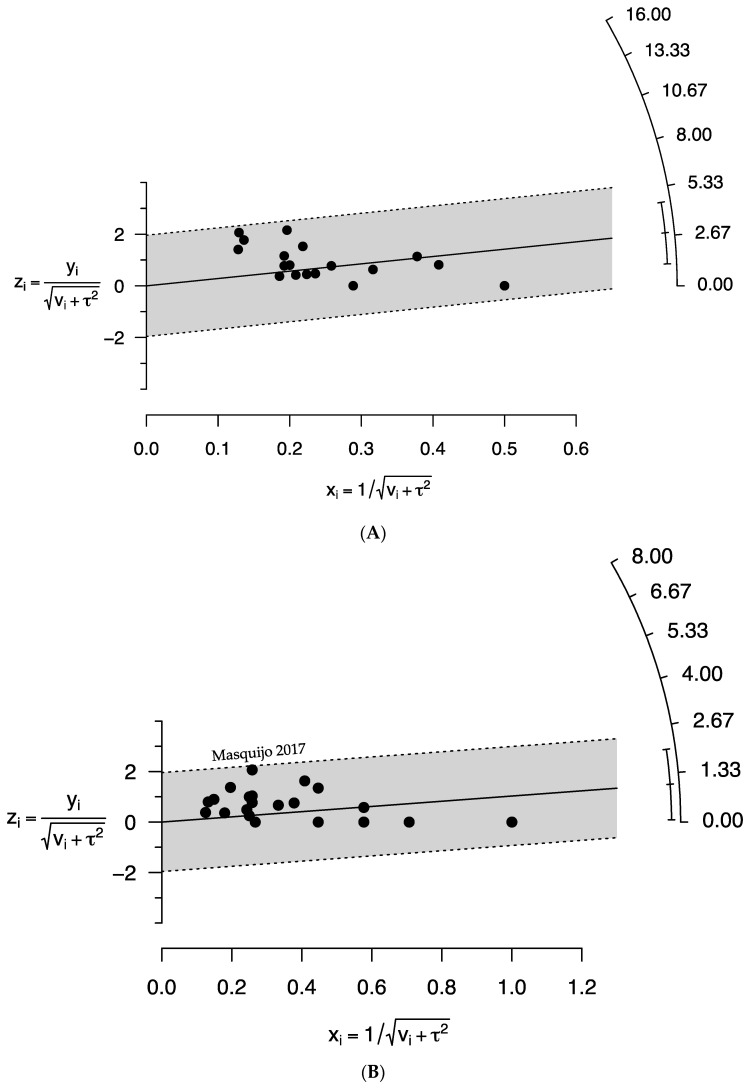
Galbraith plots representing outlier studies, if any, assessing the prevalence of avascular necrosis followed by (**A**) closed pinning and (**B**) open reduction in patients with unstable slipped capital femoral epiphysis.

**Table 2 children-09-01374-t002:** Subgroup analyses of prevalence.

Subgroups	Number of Studies Analyzed	Number of Subjects	Analyses Estimating the Prevalence	
			Prevalence of AVN [95% CI]	Heterogeneity (I^2^)
Pinning in situ	13	188	18.5% [13.3–25.1%]	0%
Pinning following intentional closed reduction	10	216	23.0% [15.3–33.0%]	44%
Pinning following unintentional closed reduction	4	58	27.6% [12.2–51.0%]	45%
Pinning following intentional and unintentional closed reduction	14	274	24.2% [17.2–33.0%]	41%
Open reduction via the Parsch method	4	79	9.9% [3.2–27.0%]	27%
Subcapital osteotomy via the anterior or anterolateral approach	7	141	18.9% [11.7–29.1%]	22%
Modified Dunn via surgical hip dislocation	14	185	19.9% [11.7–31.8%]	48%

**Table 3 children-09-01374-t003:** Number of cases and rates of AVN according to severity of slip in studies in which the Parsch method was performed.

Studies	Severity of Slip (79 Cases)				
	Mild	Moderate	Severe	AVN (%)	Total
Chen 2009	NR	NR	NR	0 (0%)	5
Herngren 2018	NR	NR	NR	1 (50%)	2
Parsch 2009	20 (31.2%) (0 AVN)	24 (37.5%) (2 AVN)	20 (31.2%) (1 AVN)	3 (5%)	64
Sankar 2010	NR	NR	NR	1 (12%)	8

NR = not reported.

**Table 4 children-09-01374-t004:** Rates of AVN according to severity and method of intervention in closed pinning.

Severity	Pinning In Situ		Pinning with Intentional Closed Reduction		Pinning with Unintentional Closed Reduction
	Cases	AVN	Cases	AVN	
Mild	26/73 (36%)	5 (19.2%)	19/185 (10%)	3 (15.7%)	No analysis, since no severity assessed in this group
Moderate	21/73 (29%)	4 (19.0%)	98/185 (53%)	16 (16.3%)	
Severe	26/73 (36%)	5 (19.2%)	68/185 (37%)	17 (25.0%)	

**Table 5 children-09-01374-t005:** Rates of AVN according to severity in open reduction.

Severity	Parsch Method		Subcapital Osteotomy		Modified Dunn	
	Cases	AVN	Cases	AVN	Cases	AVN
Mild	20/64 (31%)	0	1/108 (0.9%)	0	1/54 (1.8%)	1 (100%)
Moderate	24/64 (38%)	2 (8.3%)	0	0	5/54 (9.2%)	2 (40%)
Severe	20/64 (31%)	1 (5%)	107/108 (99.1%)	15 (14.0%)	48/54 (88.9%)	5 (10.4%)

**Table 6 children-09-01374-t006:** Sensitivity analyses.

Strategies of Sensitivity Analyses	Prevalence of AVN [95% CI] (%)	Difference in Pooled Prevalence Compared to the Main Result	Number of Studies Analyzed	Total Number of Subjects	Heterogeneity	
					*I* ^2^	*p*-Value
Closed Pinning						
Excluding low- and moderate-quality studies	24.5 [18.8–31.3]	2.6% higher	7	194	7%	0.37
Considering only cross-sectional studies	21.9 [16.3–28.7]	0.0%	15	339	32%	0.11
Excluding outlier studies	21.9 [17.5–27.2]	0.0%	19	452	26%	0.15
Open Reduction						
Excluding low- and moderate-quality studies	12.6 [8.6–18.2]	5.9% lower	9	208	0%	0.52
Considering only cross-sectional studies	16.4 [11.5–22.8]	2.1% lower	18	305	19%	0.23
Excluding outlier studies	16.8% [12.6–26.3]	1.7 lower	23	405	49%	<0.01

CI = confidence interval.

## Data Availability

Data are contained within the article or in the Appendix A.

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
