# Peer review of "Prevalence of Avascular Necrosis Following Surgical Treatments in Unstable Slipped Capital Femoral Epiphysis (SCFE): A Systematic Review and Meta-Analysis"

_children, 2022, doi:10.3390/children9091374_

Round 1

Reviewer 1 Report

Dear authors;

Thank you for the opportunity to deal with your excellent work. You comprehensively offer a differenciated overview of nowadays methods dealing with ECF.

For improvement of results presentation, please remove figures 4 and 5.

Kind Regards

Author Response

Thank you very much for your comment. I would prefer to keep the figure 4 and figure 5 in the manuscript with some justifications.

  1. Figure 4 has important message to deliver to the readers. The readers are also very much interested to know whether the review and meta-analysis have publication bias or not based on the distribution of studies on the funnel plot as well as on the value of Egger`s test.
  2. Figure 5 also has its own strength by identifying any potential source of  heterogeneity in the meta-analysis. The Galbraith plot will reveal how many studies if any, which would have contributed as outliers in the analysis. Some readers are really concerned about the numbers of outliers and the source of heterogeneity.

Reviewer 2 Report

Dear Authors: I would like to congratulate with You for Your work. In my opinion, I would better underline that more complex surgical procedure are need in more severe SCFE, and subsequently AVN cannot be correlated to the procedure itself but the stage of presentation of the pathology has to be considered as the risk factor. best regards,

Author Response

The reviewer comment and suggestion have been added in the 5th paragraph inside the discussion. The added suggestion was marked as red in the track changes in Microsoft words.

Below is the sentences (red sentences) that I added in the 5th paragraph inside the discussion.

From this meta-analysis study, as shown in Table 2, the prevalence of AVN in the Parsch method was the lowest compared to subcapital osteotomy and modified Dunn methods. However, the Parsch method should be considered with caution since the majority of the slips undergoing the Parsch method were mild and moderate types, whereas subcapital osteotomy and modified Dunn procedures dealt predominantly with severely displaced slips as in Table 6. More complex surgical procedures were needed in more severe SCFE, and subsequently AVN cannot be correlated to the procedure itself but the stage of presentation of the pathology had to be considered as the risk factor.

Reviewer 3 Report

Authors of this interesting review article  descibed and  determined   on the PubMed, Google , Scholar, Scopus and Cochrane Library databases the prevalence of femoral head of avascular necrosis   following various surgical treatments in unstable  slipped capital femoral epiphysis . It is very well and properly planeed, and organized analitic study. Statictic analysis , methodology, very readable presentation all this factors are on high scientific  level.Onlu one remark: why such commonly available databases like Publons or ResearchGate were not taken into account. To summarize : an article: "Prevalence of Avascular Necrosis Following Surgical Treatments in Unstable Slipped Capital Femoral Epiphysis (SCFE):  A Systematic Review and Meta-Analysis" is worth to be publish in Children.

Author Response

Many thanks for your comment. Neither Publons nor ResearchGate are scientific databases, rather these are social interactive platforms which is mainly based on author's personal input unlike other scientific databases like PubMed, Web of Science, Scopus and others where the journals are indexed in and the published papers are available. Publons is a commercial website that provides a free service for academics to track, verify, and showcase their peer review and editorial contributions for academic journals. Whereas, ResearchGate is a European commercial social networking site for scientists and researchers to share papers, ask and answer questions, and find collaborators. Therefore, in any systematic review, to consider Publons and Researchgate would be rather inappropriate, unfortunately. We hope it is understandable now. Thank you for your comment.